# 🍸 Mixing It Up:

# The Cocktail Effect of Multi-Task Fine-Tuning on LLM Performance - A Case Study in Finance

## Abstract

The application of large language models (LLMs) in domain-specific contexts, including finance, has expanded rapidly. Domain-specific LLMs are typically evaluated based on their performance in various downstream tasks relevant to the domain. In this work, we present a detailed analysis of fine-tuning LLMs for such tasks. Somewhat counterintuitively, we find that in domain-specific cases, fine-tuning exclusively on the target task is not always the most effective strategy. Instead, multi-task fine-tuning - where models are trained on a cocktail of related tasks - can significantly enhance performance. We demonstrate how this approach enables a small model, such as Phi-3-Mini, to achieve state-of-the-art results, even surpassing the much larger GPT-4-o model on financial benchmarks. Our study involves a large-scale experiment, conducting over 200 training experiments using several widely adopted LLMs as baselines, and empirically confirms the benefits of multi-task fine-tuning. Additionally, we explore the use of general instruction data as a form of regularization, suggesting that it helps minimize performance degradation. We also investigate the inclusion of mathematical data, finding improvements in numerical reasoning that transfer effectively to financial tasks. Finally, we note that while fine-tuning for downstream tasks leads to targeted improvements in task performance, it does not necessarily result in broader gains in domain knowledge or complex domain reasoning abilities.

## 1 Introduction

Recently, the application of large language models (LLMs) in domain-specific contexts has seen rapid growth, particularly in fields such as medicine (Singhal et al., 2023; Wu et al., 2024), law (Huang et al., 2023), and finance (Cheng et al., 2023; Wu et al., 2023). As LLMs are increasingly adopted across various domains, accurate evaluation of their domain-specific capabilities has become more necessary. While many benchmarks exist to evaluate LLM performance, they are typically designed for general purposes and not specifically for domain-specific evaluations.

A common method for assessing LLM performance within a domain is through downstream tasks (Yang et al., 2024; Gu et al., 2021; Xie et al., 2024b). Such benchmarks emphasize well-defined, highly specific tasks that seek to reflect real-world applications within the target domain. These tasks are frequently framed as standard natural language processing (NLP) problems, such as text classification, summarization, causal reasoning, arithmetic reasoning, and more. While each test individually provides limited insight into domain-specific capabilities, when combined, they offer a broader representation, facilitating a more comprehensive evaluation.

LLMs possess zero-shot capabilities (Kojima et al., 2022), allowing them to perform downstream tasks without prior task-specific training. However, they sometimes struggle with these tasks due to issues such as formatting, problem understanding, or reasoning failures. A common approach to

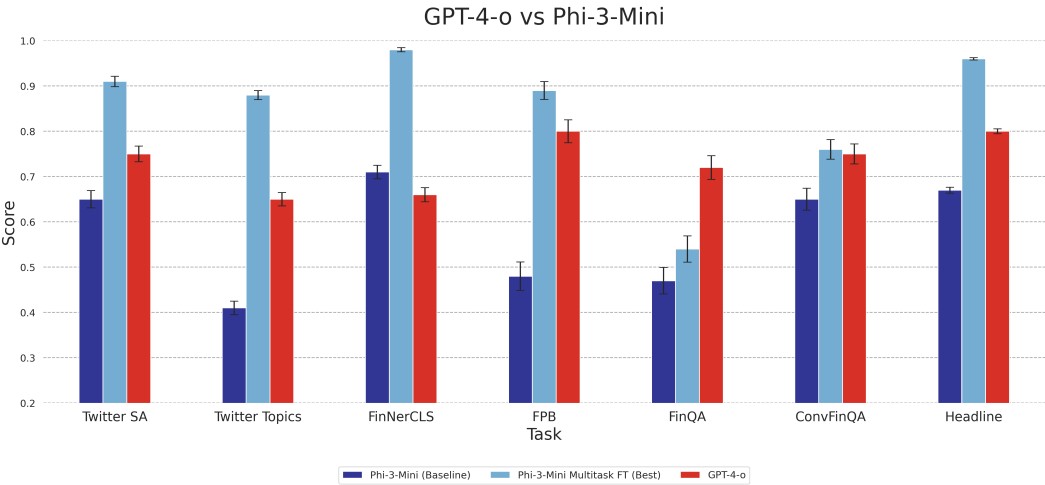

Figure 1: A comparison of performance across financial tasks between GPT-4-o, the baseline Phi-3-Mini model, and the best results achieved by multi-task fine-tuning of Phi-3-Mini.

improve their performance is to fine-tune the models directly on the downstream task, improving performance on it directly (Zhou et al., 2023). Consequently, many benchmarks provide both training and test splits to facilitate fine-tuning and evaluation. Still, fine-tuning on a single task may not fully optimize the model's performance.

In this work, we investigated the impact of *multi-task fine-tuning*. Instead of fine-tuning the model solely on the target downstream task, we fine-tune it on multiple related downstream tasks simultaneously. We conduct a massive ablation study to explore the interactions between various financial tasks and datasets. In total, we conduct 220 training experiments to provide an in-depth evaluation of different financial benchmarks and LLMs. Our empirical findings demonstrate that incorporating training data from multiple downstream tasks creates a *cocktail effect*, where the integration of multiple datasets creates a synergistic improvement in model performance, even for a single task.

Beyond task-specific data, we explore the use of general instruction-following data during the fine-tuning process and assess its impact, suggesting that it may play a regularization role. Since financial tasks often involve numerical reasoning, we also investigate the effect of incorporating general mathematical data, particularly word problems, into the training mix.

We showcase the power of the multi-task fine-tuning approach by achieving state-of-the-art results on well-established financial benchmarks. Notably, we improve the performance of the 3.8B model Phi-3-Mini (Abdin et al., 2024), enabling it to surpass the much larger and more powerful GPT-4-o model (OpenAI, 2024) in terms of benchmark accuracy, as can be seen in Fig. 1. More details are provided in Section 4.3.

Finally, after thoroughly examining how different tasks interact, we evaluate the effect of multi-task fine-tuning on extrapolation capabilities. To assess this, we test the models on domain-specific benchmarks that were not included in the training process and analyze how fine-tuning impacts performance. Our results suggest that training on downstream tasks alone may not lead to significant improvements in domain knowledge or complex reasoning abilities.

## 2 MULTI-TASK FINE-TUNING

Given a set of downstream tasks that have been selected to assess a model's capabilities in a target domain, the challenge becomes finding the optimal way to fine-tune the model across these tasks to maximize performance. In multi-task learning, the goal is to assess whether there exist synergies among the tasks, allowing for leveraging shared information to enhance individual task performance.

## 2.1 BACKGROUND

Multi-task learning is not a new concept (Caruana, 1997). The efficiency of this approach has been demonstrated across various machine learning architectures in the past (Crawshaw, 2020). This is also true for general domains in natural language processing (Aribandi et al., 2021; Aghajanyan et al., 2021; Liu et al., 2019). More recent work has shown success with instruction tuning specifically (Wang et al., 2023b; Yue et al., 2023), as well as showing the impact of additional datasets. On the other hand, the exact interactions between tasks are still understudied, especially in the domain-specific case, and more specifically for finance. Past approaches to domain-specific adaptation, such as Cheng et al. (2023), used broader domain data, removing the ability to observe the interactions between the tasks themselves. While Wang et al. (2023a) use a task oriented approach in finance, there is no measurement on the task level, or experimentation around adding general data.

## 2.2 PROBLEM FORMULATION

Let $\mathcal{M}$ be a pre-trained language model, and let $\mathcal{D} = \{D_1, D_2, \ldots, D_n\}$ represent a set of $n$ datasets used for fine-tuning. The set $\mathcal{D}$ is partitioned into two subsets: domain-specific datasets $\mathcal{D}_{\text{domain}} = \{D_1, \ldots, D_k\}$, which correspond to tasks $T_1, \ldots, T_k$, and general datasets $\mathcal{D}_{\text{gen}} = \{D_{k+1}, \ldots, D_n\}$, which are not directly evaluated in the test tasks. Our goal is to determine what is the optimal combination of datasets for fine-tuning $\mathcal{M}$ to maximize performance on a domain-specific task.

The task-level objective for multi-task fine-tuning can be formalized as:

$$\mathcal{D}_i^* = \arg\max_{\mathcal{D}_i} \left( \mathcal{E}_{T_i}(\mathcal{M}_{\mathcal{D}_i}) \right) \tag{1}$$

where $\mathcal{M}_{\mathcal{D}_i}$ represents the model trained on $\mathcal{D}_i \subseteq \mathcal{D}$, and $\mathcal{E}_{T_i}$ represents the specific evaluation metric for $T_i$.

The key questions we aim to address are:

1. Given $\mathcal{D}$, is fine-tuning on the domain-specific dataset $D_i$ *alone* the most effective way to improve performance on task $T_i$ (i.e., does $\mathcal{D}_i^* = \{D_i\}$)?

2. Can fine-tuning on general datasets $D_j \in \mathcal{D}_{\text{gen}}$ improve performance on the domain-specific tasks $T_1, \ldots, T_k$?

## 2.3 METHODOLOGY

To investigate these questions, we employ a systematic empirical approach by fine-tuning the model on different combinations of datasets. We use an incremental approach for fine-tuning the model, starting from single-dataset fine-tuning to more complex mixtures. This methodology allows us to isolate the impact of individual datasets as well as explore the interactions between datasets when fine-tuned together. All fine-tuning steps use the base model $\mathcal{M}$, and a standard uniform shuffling of $\mathcal{D}_i$. An overview of our approach for $n$ training datasets is shown in Fig. 2.

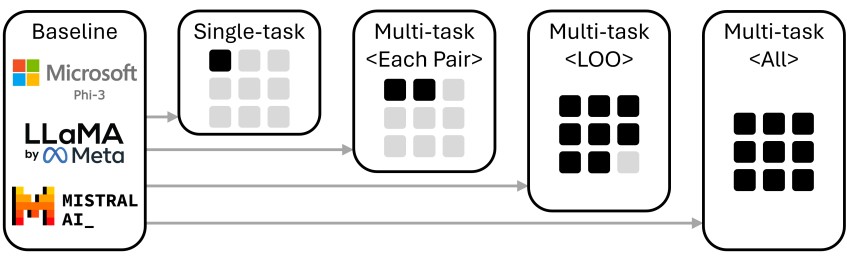

Figure 2: Overview of the methodology. The steps are: $\binom{n}{0} \rightarrow \binom{n}{1} \rightarrow \binom{n}{2} \rightarrow \binom{n}{n-1} \rightarrow \binom{n}{n}$.

Table 1: Summary statistics of the datasets used for training.

| Dataset | #Samples Train | #Samples Test | Avg. #Tokens |
|---|---|---|---|
| **Headline** | 10,000 | 20,547 | 14.8 |
| **FPB** | 3,876 | 970 | 30.3 |
| **FinNerCLS** | 5,000 | 3,502 | 62.3 |
| **FinQA** | 2,000 | 1,125 | 902.8 |
| **ConvFinQA** | 2,000 | 1,486 | 1,085.58 |
| **Twitter-Topics** | 2,500 | 4,117 | 41.9 |
| **Twitter-SA** | 5,000 | 2,388 | 25.6 |
| **Orca-Math** | 15,188 | NA | 313.5 |
| **Open-Orca** | 30,376 | NA | 340.5 |

Before fine-tuning, we evaluate the 'vanilla' model in its pre-trained state. This step establishes the baseline for all further comparisons, allowing us to quantify the relative changes in performance when fine-tuning.

After the initial fine-tuning stage, we use a single dataset at a time. We use this step primarily to understand the performance of standard single task finetuning. Additionally, this step enables us to identify the number of samples required from each dataset for stable convergence of the training loss (in less than three epochs).

To explore the interactions between datasets, we fine-tune the model on pairs of datasets. By training on two datasets simultaneously, we aim to investigate the degree of influence each dataset has on improving or impairing the model's performance on another.

Next, to fully understand the impact each dataset has, we remove a single dataset at a time, and use all other datasets for training. This step is crucial for understanding exactly how much a specific dataset influences the overall results when added to a cocktail.

Finally, we fine-tune the model on the entire set of datasets simultaneously, completing the study.

## 3 DATASETS

As part of our study we selected a variety of datasets for training and evaluation. These datasets represent central downstream NLP tasks from the financial domain, covering central benchmarks from previous works (Wu et al., 2023; Cheng et al., 2023; Wang et al., 2023a). These tasks include named entity recognition (NER), sentiment analysis, numerical reasoning, and other domain-specific challenges. The datasets are categorized into two: training and evaluation datasets. The training set includes two general datasets, as well as the training split of seven financial tasks. The evaluation set includes the test split of the seven tasks and additional datasets aimed at testing broader financial reasoning abilities. Descriptions of the datasets are below, a summary of their key properties can be found in Table 1, and an example from each dataset can be found in Appendix E.

### 3.1 CORE FINANCIAL DATASETS

The following datasets are used both for fine-tuning and for evaluation:

- **Headline**: This dataset consists of financial news headlines, accompanied by binary questions. The dataset aims to represent how financial information is presented in news media, and the primary purpose of the dataset is event detection in finance. This dataset is an adaptation of the original headline dataset (Sinha & Khandait, 2021) by FinGPT (Wang et al., 2023a).

- **FPB**: The Financial PhraseBank (FPB) (Malo et al., 2014) dataset is widely used for sentiment analysis in the financial domain. It contains annotated financial phrases and sentences, allowing the model to learn financial sentiment nuances.

- **FinNerCLS**: This dataset, created by FinGPT (Wang et al., 2023a), frames named entity recognition (NER) in finance as a classification task. This allows for more straightforward evaluation, and greater similarity to other tasks. The dataset includes sentences, entities from the sentence, and entity type labels.

- **FinQA**: FinQA (Chen et al., 2021) is a question-answering dataset that contains real-world financial documents and requires models to extract and reason over financial data to provide accurate answers. It focuses on reading comprehension tasks in finance involving numerical reasoning.

- **ConvFinQA**: The ConvFinQA dataset (Chen et al., 2022) extends FinQA by including conversational aspects, making the question-answering process more complex. It tests the model's ability to handle multi-turn financial dialogues when extracting relevant information from financial documents. For simplicity we use the BloombergGPT (Wu et al., 2023) adaptation of the dataset.

- **Twitter-Topics**: This dataset consists of finance-related topics discussed on Twitter. Each tweet needs to be classified in to one of 20 optional labels[1].

- **Twitter-SA**: A dataset of financial-sentiment annotated tweets. Each tweet needs to be classified as one of ['Bearish', 'Bullish', 'Neutral'][2].

## 3.2 General Training Datasets

Besides the financial datasets discussed earlier, we also use two non-financial training datasets. The rationale for incorporating the first dataset is the proven benefit of instruction tuning in general (Longpre et al., 2023). Additionally, since finance-related tasks often involve mathematical reasoning, we include mathematical training data to improve the model's performance in this area. Neither of these datasets are incorporated during evaluation. The datasets are as follows:

- **Open-Orca**: Open-Orca (Lian et al., 2023) is an open source recreation of the Orca (Mukherjee et al., 2023) dataset, containing diverse instructions spanning multiple keys LLM 'skills'. The dataset was created by using GPT4 and GPT3.5 to augment the FLAN collection (Longpre et al., 2023).

- **Orca-Math**: Orca-Math (Mitra et al., 2024) is a mathematical reasoning dataset that includes synthetic mathematical word problems. This dataset does not involve any domain-specific financial knowledge, but rather is used to enhance mathematical reasoning abilities.

## 3.3 Additional Evaluation Datasets

In addition to the core datasets outlined in Section 3.1, we also use **FinanceBench** (Islam et al., 2023) and **MMLU-Pro** (Wang et al., 2024) for evaluation. The FinanceBench dataset includes pairs of real-world questions about publicly traded companies, and information extracted from financial documents for answering the questions. This dataset aims to represent real-world professional use cases. MMLU-Pro contains multiple choice questions about various domains, requiring reasoning and knowledge for answering. Each question includes 10 options, reducing the probability of guessing correctly. We use only the *business* and *economics* subsets, as they are most applicable for finance.

## 4 Evaluation and Results

### 4.1 Experiment Setup

To verify that there were no biases in the results towards a particular model, we selected three of the currently top performing small models, namely Phi-3-Small[3] (Abdin et al., 2024), Llama-3.1-8B-

---

[1]https://huggingface.co/datasets/zeroshot/twitter-financial-news-topic

[2]https://huggingface.co/datasets/zeroshot/twitter-financial-news-sentiment

[3]https://huggingface.co/microsoft/Phi-3-small-128k-instruct

Instruct[4] (Dubey et al., 2024), and Mistral-7B-Instruct-v0.3[5] (Jiang et al., 2023). Additionally, to further demonstrate the effectiveness of multi-task fine-tuning, we chose a top performing miniature model, Phi-3-Mini[6] (Abdin et al., 2024). We opted for the instruct versions of each model.

All experiments were conducted using a single machine with 2 Nvidia H100 GPUs. All experiments were done using full fine-tuning of all weights in the model. We experimented with various learning rates, ranging from $3\mathrm{e}^{-6}$ to $3\mathrm{e}^{-5}$. We used three epochs for the smaller runs ($\binom{n}{1}$, $\binom{n}{2}$), and two epochs for the rest. The longest single fine-tuning experiment took under three hours to run. This choice of hyperparameters made sure that all training runs converged well, thus enabling a fair comparison. Following the process described in Section 2.2 and using the nine datasets listed in Section 3, we ended up with 55 unique training dataset mixes, resulting in 55 distinct training runs for each of the four models - yielding a total of 220 different experiments.

## 4.2 METRICS

To properly interpret our results, we aggregate the experiments and present three main metrics for each model and downstream task: single-task fine-tuning (FT), multi-task fine-tuning, and baseline scores.

For single-task fine-tuning, we evaluate the model on the test split of a specific task after being trained exclusively on the training split of that task. Using the notation from Section 2.2, the single-task score for the $i$-th dataset is defined as:

$$\text{Single-task Score} := \mathcal{E}_{T_i}(D_i) \tag{2}$$

For multi-task fine-tuning, we consider all multi-task experiments where one of the training datasets is the relevant dataset for the target task, combined with other datasets. The multi-task score is computed as:

$$\text{Multi-task Score} := \max_{\mathcal{D}_i}\left(\mathcal{E}_{T_i}\left(\mathcal{M}_{\mathcal{D}_i}\right)\right) = \mathcal{E}_{T_i}\left(\mathcal{M}_{\mathcal{D}_i^*}\right) \tag{3}$$

The baseline score represents the performance of the pre-trained model on the test split of the downstream task, without any fine-tuning. It is defined as:

$$\text{Baseline Score} := \mathcal{E}_{T_i}(\mathcal{M}) \tag{4}$$

**Numerical Evaluations:** FinQA and ConvFinQA require evaluating numerical exact match (EM) for scoring. To prevent issues stemming from rounding errors, or scale representations, we used a heuristic relaxation of exact match. We say that $x$ is *numerically same* to $y$ if for some small $\epsilon$, $y \pm \epsilon = x^n, n \in \{10^{-6}, 10^{-3}, 10^{-2}, 10^0, 10^2, 10^3, 10^6\}$. While not exhaustive, these are very common scales in finance (millions vs thousands vs billions, dollars vs cents, basis points, etc.).

**Classification:** To evaluate classification tasks we used standard (binary) accuracy scores.

**Open-End Evaluation:** Unlike the other datasets, FinanceBench contains open-end question. To properly score model responses, we used LLM-as-a-Judge (Zheng et al., 2023) for evaluation. Specifically, we used GPT-4-o as the LLM, and use the prompt in Appendix A. We consider only a strict match as correct (i.e. a score of 2), and normalize by dividing by two.

## 4.3 MAIN RESULTS

**The Cocktail Effect:** In Table 2, we present a comparison for the three LLMs using the metrics discussed above. A visualization of these results is provided in Fig. 3. It is clear that fine-tuning, whether single-task or multi-task, significantly improves performance compared to the baseline. Both fine-tuning approaches outperform the baseline across the vast majority of benchmarks, with

---

[4]https://huggingface.co/meta-llama/Meta-Llama-3.1-8B-Instruct

[5]https://huggingface.co/mistralai/Mistral-7B-Instruct-v0.3

[6]https://huggingface.co/microsoft/Phi-3-mini-128k-instruct

Table 2: Full experiment results for single-task and multi-task fine-tuning, aggregated across all experiments for three LLMs. Baseline results from the original models are provided for reference. The multi-task fine-tuning result represents the best performance across multi-task combinations. Margins of error are included for reference ($\alpha = 0.01$).

| | Phi3-Small | | | Mistral-7B-Instruct-v0.3 | | | Llama-3.1-8B-Instruct | | |
|---|---|---|---|---|---|---|---|---|---|
| | Baseline | Single-task | Multi-task | Baseline | Single-task | Multi-task | Baseline | Single-task | Multi-task |
| **Headline** | 0.67±0.009 | 0.67±0.009 | **0.96±0.004** | 0.69±0.008 | 0.67±0.009 | **0.95±0.004** | 0.53±0.009 | 0.67±0.009 | **0.95±0.004** |
| **FPB** | 0.48±0.041 | 0.86±0.029 | **0.89±0.026** | 0.78±0.034 | 0.67±0.039 | **0.89±0.026** | 0.76±0.035 | 0.82±0.032 | **0.89±0.026** |
| **FinNerCLS** | 0.71±0.02 | 0.96±0.009 | **0.98±0.006** | 0.66±0.021 | 0.97±0.007 | **0.98±0.006** | 0.54±0.022 | 0.97±0.007 | **0.99±0.004** |
| **FinQA** | 0.47±0.038 | 0.44±0.038 | **0.53±0.038** | 0.46±0.038 | 0.39±0.038 | **0.47±0.038** | **0.66±0.036** | 0.61±0.038 | 0.62±0.037 |
| **ConvFinQA** | 0.65±0.032 | 0.73±0.03 | **0.81±0.026** | 0.70±0.031 | 0.72±0.03 | **0.81±0.026** | 0.77±0.028 | 0.83±0.025 | **0.85±0.024** |
| **TwitterTopics** | 0.41±0.02 | 0.87±0.014 | **0.88±0.013** | 0.48±0.02 | 0.85±0.014 | **0.88±0.013** | 0.52±0.02 | 0.86±0.014 | **0.87±0.014** |
| **Twitter SA** | 0.65±0.025 | 0.85±0.019 | **0.91±0.015** | 0.80±0.021 | 0.83±0.02 | **0.91±0.015** | 0.68±0.025 | 0.80±0.021 | **0.91±0.015** |

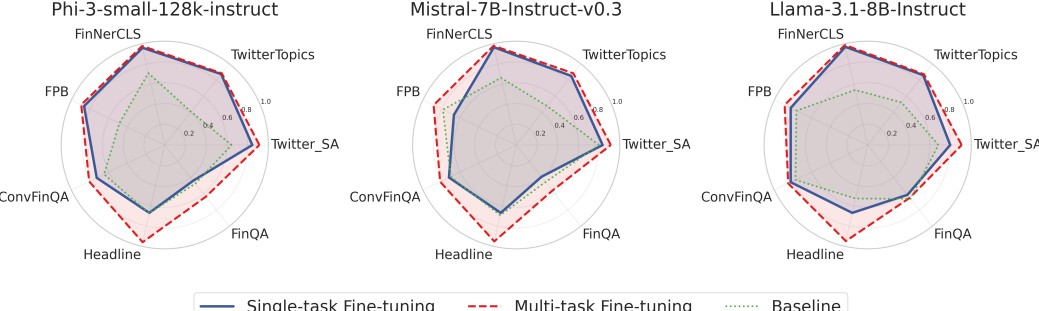

Figure 3: A visualization of Table 2. The experiment results for single-task and multi-task fine-tuning, aggregated across all experiments.

this trend holding consistently across all three models. Margins of error were calculated in the standard way, i.e. $z_{\frac{\alpha}{2}} \sqrt{\sigma^2/n}$.

When comparing multi-task and single-task performance, we observe a distinct advantage in favor of multi-task fine-tuning. Notably, there is a performance boost on the Headline and Twitter Sentiment Analysis tasks, which rely heavily on the model's ability to interpret and generate stylistically appropriate responses. The clear improvements on all tasks demonstrate the cocktail effect of multi-task fine-tuning and show the robustness of this method. Appendix D contains more in depth results regarding optimal dataset interactions, showing the top combinations per task.

**Phi-3-Mini**: To further stress-test this concept, we shifted our focus to the smaller Phi-3-Mini model, with 3.8 billion parameters, approximately 50% smaller than the primary LLMs used in our previous experiments. We replicated the same experiments but this time compared the results with the significantly larger and state-of-the-art GPT-4-o model. The results, summarized in Fig. 1, highlight a substantial performance gap between the baseline Phi-3-Mini and GPT-4-o (with the exception of the FinNerCLS task).

However, by fine-tuning the model on the datasets mentioned above, we significantly outperformed GPT-4-o on most tasks. All classification tasks showed substantial improvements over GPT-4-o, emphasizing the effectiveness of targeted fine-tuning. Notably, a fine-tuned Phi-3-Mini model even slightly outperformed GPT-4-o on the challenging ConvFinQA benchmark. ConvFinQA involves conversations, which likely provide implicit few-shot learning opportunities, enabling the model to better understand and anticipate the structure of the questions. This contrasts with the FinQA dataset, which lacks conversational context, resulting in only modest gains for the fine-tuned model.

Table 3: Performance comparison for MMLU-Pro Business, MMLU-Pro Economics, and FinanceBench. For each model the best multi-task fine-tuning score is compared with the baseline.

| | MMLU-Pro Business | | MMLU-Pro Economics | | FinanceBench | |
|---|---|---|---|---|---|---|
| | Baseline | Multi-task | Baseline | Multi-task | Baseline | Multi-task |
| **Mistral-7B-Instruct-v0.3** | **0.3207** | 0.2548 | **0.4716** | 0.4040 | 0.4533 | **0.4667** |
| **Llama-3.1-8B-Instruct** | **0.5296** | 0.4068 | 0.4716 | **0.5213** | 0.6133 | **0.6733** |
| **Phi-3-Mini** | **0.4702** | 0.3904 | **0.6149** | 0.5652 | **0.4733** | 0.4667 |
| **Phi-3-Small-128k-instruct** | **0.5361** | 0.4461 | **0.6647** | 0.6078 | 0.5867 | **0.6400** |

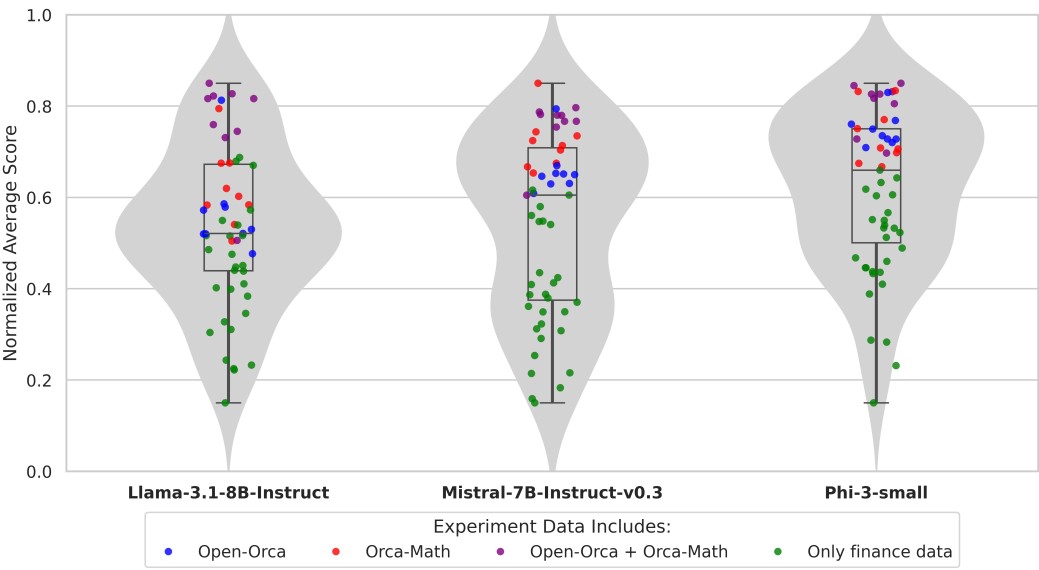

Figure 4: Normalized averaged scores for all seven core tasks described in Section 3.1 across all experiments. Each point represents the average score for a single fine-tuned model. The colors represent the type of datasets used in the experiment.

This experiment demonstrates that by using multi-task fine-tuning, and by specifically targeting downstream tasks, it is possible to outperform much larger and more powerful models in these tasks. The full results are presented in Appendix B.

**Domain Generalization** With the exception of Llama on FinQA, all the downstream tasks improve significantly with multi-task finetuning, across all models. Table 3 shows that this trend does not necessarily implicate that the models have improved in the general finance domain. While there may be some improvement in FinanceBench, there is no clear improvement in the other two tasks, and possibly even a regression. This finding raises a strong concern regarding the use of these downstream tasks, or many of the other commonly used benchmarks, as proxies for successful domain adaptation.

**Data Regularization Hypothesis** We provide a further analysis of the data by examining the effect of the two non-financial datasets: Open-Orca and Orca-Math. In Fig. 4 we present a summary of all fine-tuning experiments. We compute the average score of each fine-tuned model across the seven core tasks described in Section 3.1. For visualization purposes, we normalize the results for each model separately to be between $0.15$ and $0.85$. There is a clear distinction between models that used the non-financial datasets, and models that relied purely on the downstream tasks.

Open-Orca performs well across tasks and models. Unlike Orca-math, where strengthening mathematical reasoning abilities is directly related to model performance on tasks, it is nontrivial to interpret why adding general data would help with domain-specific downstream tasks. Moreover, it is very likely that the models were exposed to this data during pre-training, i.e., no new reasoning abilities were added.

When aligning LLMs, Ouyang et al. (2022) adapt the loss used by Stiennon et al. (2020), including a regularization term: $\beta \log\left[\mathcal{M}_{\text{RL},\phi}(y|x)/\mathcal{M}_{\text{SFT}}(y|x)\right]$. This component is used to ensure the new model does not stray 'too far' from the original model, and is missing in the standard domain adaptation regime. We hypothesize that since the pretrained model $\mathcal{M}$ has already been exposed to Open-Orca, incorporating it in finetuning serves a similar purpose. In other words, we assume:

$$\log\left[\mathcal{M}_{\mathcal{D}_{\text{domain}}}(y|x)/\mathcal{M}(y|x)\right] \geq \log\left[\mathcal{M}_{(\mathcal{D}_{\text{domain}}\cup\mathcal{D}_{\text{gen}})}(y|x)/\mathcal{M}(y|x)\right].$$

We leave the exploration and research of this hypothesis to future work.

## 5 RELATED WORK

**Domain-specific LLMs:** Recent advances in LLMs have led to many attempts at creating models tailored to specific domains. These models aim to outperform general-purpose ones by having deeper knowledge of the domain, being more effective at solving tasks relevant to that domain, or adopting a more appropriate style. Several methods have been suggested for training these models. One approach is to pre-train a language model entirely on domain-specific data, as seen in (Wu et al., 2023; Singhal et al., 2023). Another common approach is to take pre-trained LLMs and fine-tune them for specific downstream tasks (Xie et al., 2023b; Wang et al., 2023a; Cheng et al., 2024; Jiang et al., 2024; Cheng et al., 2023) in a domain adaptation process.

**Domain Adaptation of LLMs:** Various techniques have been developed to transform a general language model into a domain-specific one. One option is continual pre-training (CPT) (Gururangan et al., 2020), where a pre-trained LLM undergoes further training on raw data that contains relevant domain-specific knowledge, enhancing the model's understanding of that domain. Another method involves supervised fine-tuning (SFT), where the model is trained on a large set of domain-specific instructions (Wei et al., 2021). Some approaches focus on specific tasks within the domain, fine-tuning the model with instruction datasets tailored to those particular tasks (Wang et al., 2023a). There are also various works on approaches for selecting data for training (Xie et al., 2023a; Xia et al., 2024).Additionally, a hybrid approach has been proposed, where CPT is performed first, followed by domain-specific instruction tuning to refine the model's capabilities (Bhatia et al., 2024; Wu et al., 2024; Xie et al., 2024b;c).

**Finance Benchmarks:** With the increasing adoption of LLMs, several benchmarks have been proposed to evaluate model performance in the financial domain. Recently, efforts have been made to combine existing tests and datasets into more comprehensive evaluation frameworks. For instance, FinBen (Xie et al., 2024a), PIXIU (Xie et al., 2024b), and BBT-Fin (Lu et al., 2023) aggregate a variety of common tasks to provide a broad analysis of general financial skills. Other benchmarks focus on more specialized scenarios. For example, FinEval (Zhang et al., 2023) was developed to assess LLM financial knowledge based on academic textbooks, while SuperCLUE-Fin (Xu et al., 2024) aims to replicate real-world financial tasks through a detailed breakdown of subtasks. Another example is FinDABench (Liu et al., 2024), which places a strong emphasis on financial analysis and reasoning rather than pure knowledge evaluation.

## 6 CONCLUSIONS

In this work, we demonstrated the potential of multi-task fine-tuning as a robust approach to optimizing the performance of LLMs on downstream tasks. Through extensive experimentation involving over 200 training runs, we showed that combining training data from multiple related financial tasks creates a "cocktail effect", yielding significant performance gains, and even allowing smaller models such as Phi-3-Mini to surpass larger counterparts like GPT-4-o on targeted benchmarks. Our findings highlight the advantages of a training approach that leverages synergies between tasks.

Furthermore, our exploration of integrating general instruction-following and mathematical datasets demonstrated promising results, combining what may be a regularization effect, with an enhance-

ment of numerical reasoning abilities. Nevertheless, we observed that while multi-task fine-tuning significantly boosts specific task performance, it does not necessarily translate into improved overall domain knowledge. This suggests that while multi-task fine-tuning is effective for task-specific improvements, broader gains in domain competency may require more sophisticated strategies.

Overall, our results provide strong empirical evidence for the benefits of multi-task fine-tuning in domain-specific model adaptation. This approach not only optimizes task performance but also underscores the importance of thoughtful dataset selection and the value of leveraging cross-task learning. Future work may benefit from exploring hybrid approaches that combine multi-task learning with targeted domain adaptation, aiming to bridge the gap between task-specific proficiency and more comprehensive domain understanding.

### LIMITATIONS

We acknowledge several limitations of this work. As with all experiments involving fine-tuning, the choice of hyperparameters plays a critical role. While we conducted a targeted hyperparameter search, the large scale of our experiments made a comprehensive grid search infeasible.

Additionally, the financial domain is vast, encompassing many intricacies and complexities that extend beyond the scope of the seven core datasets used in this study. Our work serves as a case study focusing on these representative datasets, but addressing other aspects of finance will necessitate the use of additional datasets tailored to those specific areas.

Finally, we note that while there are plenty of empirical results that demonstrate the general effectiveness of multi-task learning, there is still a significant lack of modern theory (Crawshaw, 2020). Although past works provide strong theoretical frameworks for multi-task learning (Evgeniou & Pontil, 2004; Ciliberto et al., 2015), it is difficult to extend them elegantly to modern deep learning methods.

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

## A    LLM as a Judge Prompt

We used the following prompt:

> <Instruction >
> Please act as an impartial judge and evaluate the quality of the response provided by an AI assistant to the user question displayed below. You will be given a reference answer and the assistant's answer. Begin your evaluation by comparing the assistant's answer with the reference answer. Identify and correct any mistakes. Be as objective as possible. After providing your explanation, you must rate the response on a scale of 0 to 2 by strictly following this format: [[rating]], for example: The rating is: [[1]], or: My rating is [[0]].
> Note! The answers have to answer the question correctly, but they do not have to be identical, or equally detailed, or equally helpful! You are only measuring

equality of correctness, not completeness. Be forgiving of rounding errors, as long as they are not essential, as well as over/under explaining.

You should provide a 0 rating when the answers does not match the reference.

You should provide a 1 rating when the answer is partially correct.

You should provide a 2 rating when the answer is correct.

For example, if the reference answer is "It cost $5B annually" and the assistant answer is "It cost $5 billion per year", the rating should be 2.

If the assistant answer is "It cost $5", the rating should be 1.

If the assistant answer is "It cost $4 million per month", the rating should be 0.

For example, if the reference answer is a list of most major locations on Earth and the assistant replies concisely 'Globally', the rating should be 2.

If the assistant replies 'A variety of places worldwide', the rating should be 1.

If the assistant replies 'In Europe', the rating should be 0.

For example, if the question is "What was his salary?" and the reference answer is "We can see that by adding the various components in table 3, we get that 3K + 7.5K equals a total salary of 10.5K annually", and the assistant's answer is "10,500", the rating should be 2.

If the assistant's answer is "10.5K. This salary reflects and excellent compensation given the low cost of living in the area", the rating should still be 2.

If the assistant's answer is "the answer can be found in table 3 by adding 3K + 7.5K", the rating should be 1.

If the assistant's answer is "7.5K", the rating should be 0.

$</$Instruction $>$

$<$Question $>$

{question}

$</$Question $>$

$<$Reference Answer $>$

{ref_answer}

$</$Reference Answer $>$

$<$Assistant's Answer $>$

{answer}

$</$Assistant's Answer $>$

## B    PHI-3-MINI FULL RESULTS

Table 4: Comparison of GPT-4-o to Phi-3-Mini, including its baseline, single-task fine-tuning, and multi-task fine-tuning variants.

| | Phi-3-Mini | | | GPT-4-o |
|---|---|---|---|---|
| | Baseline | Single-task FT | Multi-task FT | |
| Twitter SA | 0.65 | 0.66 | **0.91** | 0.75 |
| Twitter Topics | 0.41 | 0.87 | **0.88** | 0.65 |
| FinNerCLS | 0.71 | 0.97 | **0.98** | 0.66 |
| FPB | 0.48 | 0.13 | **0.89** | 0.80 |
| FinQA | 0.47 | 0.31 | 0.54 | **0.72** |
| ConvFinQA | 0.65 | 0.66 | **0.76** | 0.75 |
| Headline | 0.67 | 0.67 | **0.96** | 0.80 |

## C    FULL RESULTS

Fig. 5 is a visualization of the results from Table 2, and shows the full results for each model across all seven tasks. Phi-3-Mini is brought here as well for completeness.

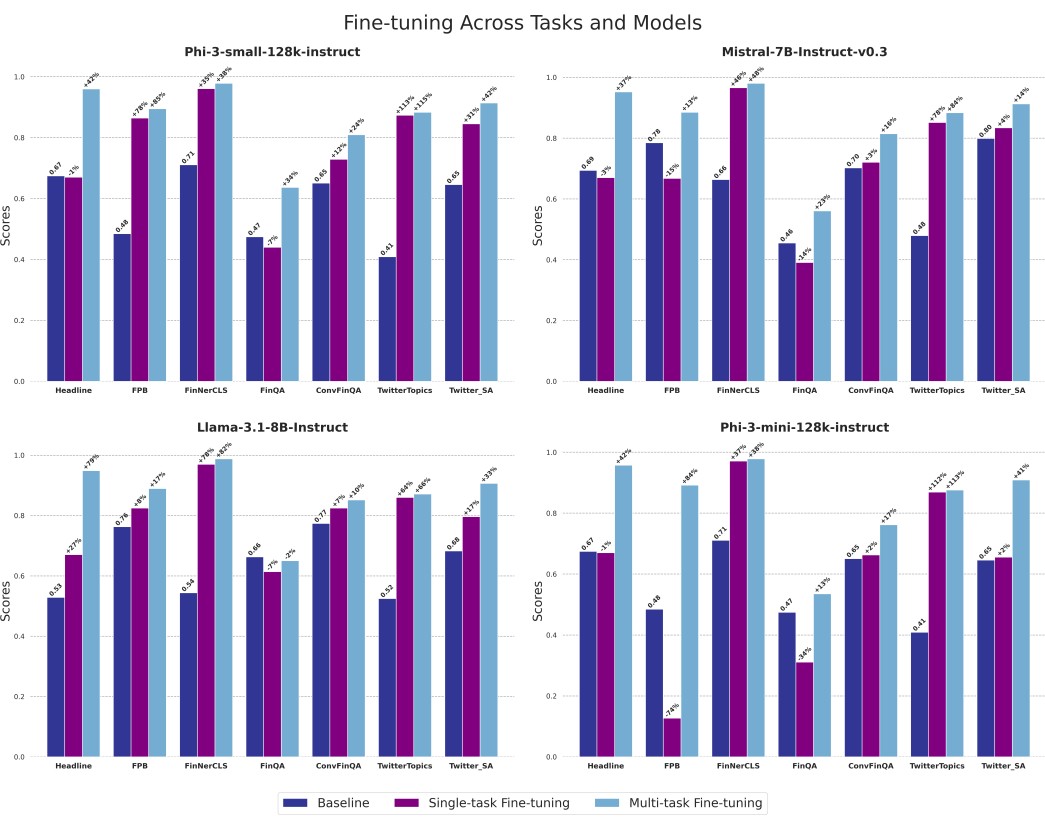

Figure 5: Evaluation scores of all four models on all seven core tasks described in Section 3.1. The relative gain (in percentage) is reported of each fine-tuning experiment.

## D    ABLATION STUDY RESULTS

Table 5, Table 6, and Table 7 present the top 3 most helpful dataset combination for Llama-3.1-8B-Instruct, Mistral-7B-Instruct-v0.3, and Phi-3-Small, respectively, across each task used in our ablation study. The tables provide detailed results for each task, showing the score achieved, the difference from the maximum score, and the percentage of the maximum score. Note that since using the dataset itself trivially enhances abilities, we only include $\mathcal{D}_i$ such that $D_i \notin \mathcal{D}_i$.

## E    DATASET EXAMPLES

DATASET: HEADLINE

**Instruction:**
Assess if the news headline touches on price in the past. Options: Yes, No
**Input:**
*april gold down 20 cents to settle at $1,116.10/oz*
**Output:**
No

Table 5: Top 3 most helpful datasets for Llama-3.1-8B-Instruct

| Task | Datasets | Score | Diff from Max | % of Max |
|---|---|---|---|---|
| Twitter_SA | Orca-Math, Headline, FPB, FinNerCLS, ConvFinQA, FinQA, TwitterTopics, Open-Orca | 0.8652 | 0.0419 | 95.38 |
| Twitter_SA | Headline, FPB | 0.8635 | 0.0436 | 95.20 |
| Twitter_SA | FPB, Open-Orca | 0.8425 | 0.0645 | 92.89 |
| TwitterTopics | FPB, Twitter_SA | 0.5903 | 0.2812 | 67.73 |
| TwitterTopics | FinNerCLS, Twitter_SA | 0.5834 | 0.2880 | 66.95 |
| TwitterTopics | FinQA, Twitter_SA | 0.5799 | 0.2915 | 66.54 |
| FinNerCLS | Headline, Open-Orca | 0.6912 | 0.2972 | 69.93 |
| FinNerCLS | Orca-Math, Open-Orca | 0.6851 | 0.3032 | 69.32 |
| FinNerCLS | ConvFinQA | 0.6805 | 0.3079 | 68.85 |
| FPB | FinQA, TwitterTopics | 0.8121 | 0.0775 | 91.29 |
| FPB | Headline, TwitterTopics | 0.8106 | 0.0791 | 91.11 |
| FPB | Twitter_SA, Open-Orca | 0.8079 | 0.0817 | 90.81 |
| ConvFinQA | FPB | 0.7927 | 0.0592 | 93.05 |
| ConvFinQA | TwitterTopics, Twitter_SA | 0.7672 | 0.0848 | 90.05 |
| ConvFinQA | FPB, FinQA | 0.7618 | 0.0902 | 89.42 |
| Headline | FPB, FinQA | 0.7235 | 0.2256 | 76.23 |
| Headline | FPB | 0.6917 | 0.2574 | 72.88 |
| Headline | FPB, FinNerCLS | 0.6899 | 0.2592 | 72.69 |
| FinQA | Orca-Math, FPB | 0.6507 | 0.0000 | 100.00 |
| FinQA | Orca-Math, TwitterTopics | 0.6480 | 0.0027 | 99.59 |
| FinQA | Orca-Math | 0.6418 | 0.0089 | 98.63 |

Table 6: Top 3 most helpful datasets for Mistral-7B-Instruct-v0.3

| Task | Datasets | Score | Diff from Max | % of Max |
|---|---|---|---|---|
| Twitter_SA | Orca-Math, Headline, FPB, FinNerCLS, ConvFinQA, FinQA, TwitterTopics, Open-Orca | 0.8643 | 0.0486 | 94.68 |
| Twitter_SA | FPB, Open-Orca | 0.8555 | 0.0574 | 93.72 |
| Twitter_SA | TwitterTopics, Open-Orca | 0.8513 | 0.0616 | 93.26 |
| TwitterTopics | Orca-Math, Headline, FPB, FinNerCLS, ConvFinQA, FinQA, Twitter_SA, Open-Orca | 0.4873 | 0.3964 | 55.14 |
| TwitterTopics | Headline, FinQA | 0.4800 | 0.4038 | 54.31 |
| TwitterTopics | FPB, Open-Orca | 0.4753 | 0.4084 | 53.78 |
| FinNerCLS | Headline, ConvFinQA | 0.7581 | 0.2226 | 77.30 |
| FinNerCLS | Headline, FinQA | 0.7353 | 0.2454 | 74.98 |
| FinNerCLS | ConvFinQA, FinQA | 0.7327 | 0.2480 | 74.72 |
| FPB | Orca-Math, Headline, FinNerCLS, ConvFinQA, FinQA, TwitterTopics, Twitter_SA, Open-Orca | 0.8193 | 0.0660 | 92.54 |
| FPB | Orca-Math, FinQA | 0.8098 | 0.0756 | 91.46 |
| FPB | Twitter_SA, Open-Orca | 0.8092 | 0.0761 | 91.40 |
| ConvFinQA | Orca-Math, FPB | 0.6891 | 0.1258 | 84.56 |
| ConvFinQA | Orca-Math | 0.6884 | 0.1265 | 84.48 |
| ConvFinQA | Orca-Math, Headline | 0.6824 | 0.1326 | 83.73 |
| Headline | TwitterTopics, Open-Orca | 0.7377 | 0.2145 | 77.48 |
| Headline | Open-Orca | 0.7299 | 0.2223 | 76.65 |
| Headline | ConvFinQA, Open-Orca | 0.7275 | 0.2247 | 76.40 |
| FinQA | Orca-Math, FPB | 0.5609 | 0.0000 | 100.00 |
| FinQA | Orca-Math, TwitterTopics | 0.5564 | 0.0044 | 99.21 |
| FinQA | Orca-Math | 0.5538 | 0.0071 | 98.73 |

DATASET: FPB

**Instruction:**
You are given a financial document. Your task is to infer its sentiment. Answer using one of the following labels: ['Negative', 'Neutral', 'Positive'], and include nothing else. You must answer with a single word, and no additional context.
**Input:**
*Under the terms of the agreement, Bunge will acquire Raisio's Keiju, Makuisa and Pyszny Duet brands and manufacturing plants in Finland and Poland.*

Table 7: Top 3 most helpful datasets for Phi-3-Small

| Task | Datasets | Score | Diff from Max | % of Max |
|------|----------|-------|---------------|----------|
| Twitter_SA | Headline, Open-Orca | 0.8677 | 0.0461 | 94.96 |
| Twitter_SA | Orca-Math, TwitterTopics | 0.8597 | 0.0540 | 94.09 |
| Twitter_SA | TwitterTopics, Open-Orca | 0.8526 | 0.0611 | 93.31 |
| TwitterTopics | Orca-Math, Headline, FPB, FinNerCLS, ConvFinQA, FinQA, Twitter_SA, Open-Orca | 0.5629 | 0.3203 | 63.74 |
| TwitterTopics | Headline, Open-Orca | 0.5449 | 0.3383 | 61.70 |
| TwitterTopics | ConvFinQA, Open-Orca | 0.5418 | 0.3414 | 61.34 |
| FinNerCLS | Orca-Math, ConvFinQA | 0.7912 | 0.1872 | 80.87 |
| FinNerCLS | ConvFinQA, Open-Orca | 0.7866 | 0.1919 | 80.39 |
| FinNerCLS | Orca-Math, FinQA | 0.7702 | 0.2082 | 78.72 |
| FPB | Orca-Math, Headline, FinNerCLS, ConvFinQA, FinQA, TwitterTopics, Twitter_SA, Open-Orca | 0.8365 | 0.0583 | 93.48 |
| FPB | Twitter_SA, Open-Orca | 0.8333 | 0.0616 | 93.12 |
| FPB | Headline, Open-Orca | 0.8189 | 0.0760 | 91.51 |
| ConvFinQA | Orca-Math, FinNerCLS | 0.7416 | 0.0680 | 91.60 |
| ConvFinQA | Orca-Math, TwitterTopics | 0.7409 | 0.0686 | 91.52 |
| ConvFinQA | Orca-Math, FPB | 0.7396 | 0.0700 | 91.35 |
| Headline | ConvFinQA, Open-Orca | 0.6956 | 0.2644 | 72.46 |
| Headline | Open-Orca | 0.6846 | 0.2754 | 71.32 |
| Headline | Orca-Math, Open-Orca | 0.6794 | 0.2806 | 70.77 |
| FinQA | Orca-Math, FinNerCLS | 0.6364 | 0.0000 | 100.00 |
| FinQA | Orca-Math, TwitterTopics | 0.6329 | 0.0036 | 99.44 |
| FinQA | Orca-Math, FPB | 0.6178 | 0.0187 | 97.07 |

**Output:**
neutral

DATASET: FINNERCLS

**Instruction:**
What is the entity type of '40 William St' in the input sentence. Options: person, location, organization

**Input:**
*This LOAN AND SECURITY AGREEMENT dated January 27, 1999, between SILICON VALLEY BANK ("Bank"), a California-chartered bank with its principal place of business at 3003 Tasman Drive, Santa Clara, California 95054 with a loan production office located at 40 William St., Ste.*

**Output:**
location

DATASET: FINQA

**Instruction:**
Please answer the given financial question based on the context.

**Input:**
*Interest rate to a variable interest rate based on the three-month LIBOR plus 2.05% (2.34% as of October 31, 2009). If LIBOR changes by 100 basis points, our annual interest expense would change by $3.8 million...*

**Question:**
What is the interest expense in 2009?

**Output:**
3.8

DATASET: CONVFINQA

**Instruction:**
Read the following texts and table with financial data from an S&P 500 earnings report carefully.

Based on the question-answer history (if provided), answer the last question. The answer may require mathematical calculation based on the data provided.

**Input:**
*Charges during the years then ended are presented below: The fair value of restricted stock that*

| - | | 2013 | 2012 | 2011 |
|---|---|---|---|---|
| 1 | balance at beginning of year | 2,804,901 | 2,912,456 | 2,728,290 |
| 2 | granted | 192,563 | 92,729 | 185,333 |
| 3 | cancelled | -3,267 | -200,284 | -1,167 |
| 4 | balance at end of year | 2,994,197 | 2,804,901 | 2,912,456 |
| 5 | vested during the year | 21,074 | 408,800 | 66,299 |
| 6 | compensation expense recorded | $6,713,155 | $6,930,381 | $17,365,401 |
| 7 | weighted average fair value of restricted stock granted during the year | $17,386,949 | $7,023,942 | $21,768,084 |

*vested during the years ended December 31, 2013, 2012, and 2011 was $1.6 million, $22.4 million, and $4.3 million, respectively.*

*Substantially in accordance with the original terms of the program, 50% of these LTIP units vested on December 17, 2012 (accelerated from the original January 1, 2013 vesting date), 25% vested on December 11, 2013 (accelerated from the original January 1, 2014 vesting date), and the remainder is scheduled to vest on January 1, 2015.*

*Question:*
What was the total, in millions, capitalized to assets associated with compensation expense related to long-term compensation plans, restricted stock, and stock options in the year of 2013?
*Output:*
4.5

**Question:**
And what was it in 2012, also in millions?
**Output:**
4.1

DATASET: TWITTER-TOPICS

**Instruction:**
You are given a financial sentence taken from Twitter. Your task is to infer its topic. Answer using one of the following labels: ['Analyst Update', 'Fed - Central Banks', 'Company - Product News', 'Treasuries - Corporate Debt', 'Dividend', 'Stock Commentary', 'Stock Movement'], and include nothing else.
**Input:**
*Here are Thursday's biggest analyst calls: Apple, Amazon, Tesla, Palantir, DocuSign, Exxon & more...*
**Output:**
Analyst Update

DATASET: TWITTER-SA

**Instruction:**
You are given a financial sentence taken from Twitter. Your task is to infer its sentiment. Answer using one of the following labels: ['Bearish', 'Bullish', 'Neutral'], and include nothing else.
**Input:**
*$BYND - JPMorgan reels in expectations on Beyond Meat...*
**Output:**
Bearish

DATASET: ORCA-MATH

**Instruction:**
You are given a mathematical word problem. Solve it step by step and provide the answer.
**Input:**
*You wanted to subtract 46 from a number, but you accidentally subtracted 59 and got 43. How much do you get from the correct calculation?*
**Output:**
If you subtract 46 from the original number, you would get 56.

DATASET: OPEN-ORCA

**Instruction:**
You are an AI assistant. You will be given a task. You must generate a detailed and long answer.
**Input:**
*Generate a fifteen-word sentence that describes all this data: Midsummer House eatType restaurant; Midsummer House food Chinese; Midsummer House priceRange moderate; Midsummer House customer rating 3 out of 5; Midsummer House near All Bar One.*
**Output:**
Midsummer House is a moderately priced Chinese restaurant with a 3/5 customer rating, located near All Bar One.

DATASET: FINANCEBENCH

**Instruction:**
You are given a financial question and a financial document. Your task is to answer the question based on the document.
**Input:**
*Amcor plc and Subsidiaries Consolidated Statements of Income ($ in millions, except per share data)...*
**Question:**
Does AMCOR have an improving gross margin profile as of FY2023? If not, explain why.
**Output:**
No. For AMCOR, there has been a slight decline in gross margins by 0.8%.

DATASET: MMLU-PRO ECONOMICS

**Instruction:**
The following are multiple choice questions (with answers) about economics. Think step by step and then finish your answer with "the answer is (X)" where X is the correct letter choice.
**Input:**
*Mr. Jones is president of the First National Bank of St. Louis and wishes to determine if his bank is holding too much of its demand deposits as reserves. The bank's total deposits = $1,700,000 and the reserve ratio is 20%. If Mr. Jones finds that reserves = $850,000 what might he conclude about excess reserves?* **Options:** *A: "$340,000", B: "$600,000", C: "$425,000", D: "25%", E: "10%", F: "$510,000", G: "$1,700,000", H: "30%", I: "$255,000", J: "15%"*
**Output:**
F

DATASET: MMLU-PRO BUSINESS

**Instruction:**
The following are multiple choice questions (with answers) about business. Think step by step and then finish your answer with "the answer is (X)" where X is the correct letter choice.
**Input:**
*Mr. Frankel wants to borrow $2,000 from November 16 for 143 days. The interest rate is 6%. What would the difference in the interest charge amount to if the bank used exact interest instead of bankers' interest?* **Options:** *A: "$2.00", B: "$0.25", C: "$1.50", D: "$1.32", E: "$3.30", F:*

*"$0.50", G: "$0.99", H: "$0.66", I: "$1.98", J: "$2.64"*
**Output:**
H

