# OpenReview forum: "Mixing It Up:  The Cocktail Effect of Multi-Task Fine-Tuning on LLM Performance - A Case Study in Finance"
_ICLR.cc/2025/Conference — ICLR 2025 Conference Withdrawn Submission_

### Official Review · Reviewer_FeNh · 2024-11-01

**Soundness:** 3
**Presentation:** 3
**Contribution:** 2
**Rating:** 5
**Confidence:** 3

**Summary:**

This paper presents a detailed analysis of fine-tuning large language models (LLMs) for domain-specific tasks, with a focus on finance. The authors propose a multi-task fine-tuning approach, where models are trained on a combination of related tasks, and demonstrate its effectiveness by achieving state-of-the-art results on financial benchmarks. They compare the performance of a smaller model, Phi-3-Mini, with a larger model, GPT-4-o, and show that multi-task fine-tuning allows Phi-3-Mini to surpass GPT-4-o. The authors also explore the use of general instruction-following and mathematical data during fine-tuning and analyze the impact of different datasets on model performance.

**Strengths:**

1. The paper presents a thorough analysis of multi-task fine-tuning for domain-specific LLMs, including an extensive experimental setup involving over 200 models.
2.  The authors provide strong empirical evidence for the benefits of multi-task fine-tuning, demonstrating significant performance gains on financial benchmarks.
3. The paper explores the use of general instruction-following and mathematical data during fine-tuning, which adds valuable insights into the regularization and numerical reasoning abilities of the models.
4. The authors use an incremental approach for fine-tuning the model, which allows them to isolate the impact of individual datasets and explore the interactions between datasets when fine-tuned together.

**Weaknesses:**

1.While the paper provides strong empirical evidence, it lacks theoretical insights into why multi-task fine-tuning is effective.
2.The paper only compares multi-task fine-tuning with single-task fine-tuning and does not consider other fine-tuning strategies such as adapter-based fine-tuning or prompt-tuning. The proposed finetuning method is similar to multi-task deep neural networks (MT-DNN), see Liu et al. 2019, ACL.

**Questions:**

See weaknesses

---

> ### Author Response · Authors · 2024-11-15
> **Response to Reviewer FeNh**
>
> We thank the reviewer for appreciating our empirical work and for raising insightful questions. We respond to all points below.
>
> **Weakness I**: We acknowledge the reviewer's point about the lack of theoretical insights. While there is substantial empirical evidence demonstrating the effectiveness of multi-task learning, modern theoretical understanding remains limited [1]. Although earlier works provide strong theoretical frameworks for multi-task learning [2,3], extending these frameworks to modern deep learning methods presents significant challenges. To address this point, we have added relevant citations to the manuscript, along with a paragraph in the limitations section.
>
> [1]: Michael Crawshaw. Multi-task learning with deep neural networks: A survey. arXiv preprint, arXiv:2009.09796,2020.
>
> [2]: Theodoros Evgeniou and Massimiliano Pontil.Regularized multi–task learning. In Proceedings of the tenth ACM SIGKDD international conference on Knowledge discovery and data mining, pp. 109–117,2004.
>
> [3]: Carlo Ciliberto, Youssef Mroueh, Tomaso Poggio, and Lorenzo Rosasco. Convex learning of multiple tasks and their structure. In International Conference on Machine Learning, pp.1548–1557. PMLR,2015.
>
> **Weakness II**:  Regarding the use of adapters and prompt-tuning, we agree with the reviewers' suggestion that exploring additional fine-tuning techniques could provide valuable insights. However, such experiments fall outside the scope of this work. Our study specifically examines different data mixes, isolating this variable to analyze its effects comprehensively. While we anticipate similar findings given the strong dependency of fine-tuned models on their training data, investigating the impact of alternative fine-tuning techniques beyond standard training of the full model and all its weights would be an interesting direction for future research.
>
> Finally, we thank the reviewer for bringing the paper by Liu et al. to our attention. While their approach is certainly related to ours, there is a major difference: rather than creating a single model capable of solving multiple tasks, they share the earlier layers of the model across tasks while using separate 'heads' for each task. This approach is understandable given the limitations of BERT-related architectures. With more modern and powerful architectures, these separate 'heads' are no longer necessary. Nonetheless, the work by Liu et al. represents an important early contribution to multitask fine-tuning, and we have added a citation to acknowledge it.

---

### Official Review · Reviewer_KnZt · 2024-11-01

**Soundness:** 2
**Presentation:** 2
**Contribution:** 2
**Rating:** 3
**Confidence:** 3

**Summary:**

In this paper, the authors investigate the effectiveness of multi-task fine-tuning as a method to enhance the performance of large language models (LLMs) on downstream tasks.

They claimed that they conduct extensive experiments with over 200 models and demonstrate that combining training data from multiple related financial tasks results in a "cocktail effect," which significantly boosts performance. They argue that their approach even enables smaller models, such as Phi-3-Mini, to outperform larger counterparts like GPT-4-o in specific benchmarks. The authors also state the benefits of this training strategy, emphasizing the synergistic advantages gained from leveraging inter-task relationships.

**Strengths:**

The paper is well-written, featuring a clear structure and logical flow, although it potentially omits some important details, as noted in the Weaknesses section.

The evaluation metrics established by the authors for different types are reasonable and appropriate but could be more comprehensive.

The presentation of visualizations is informative and insightful.

**Weaknesses:**

1. Given the author mention multiple times in abstract, introduction and conclusion sections that they have trained 200+ models, however, there is no relevant details about it in the main body of the paper. It would be necessary to include critical information such that what is the full implemented model list and how the model selection is conducted as the author seem to eventually choose Phi-3 Mini to present their research outcomes.

2. Although the authors fine-tuned their models on multi-purpose datasets, they selected a dataset with limited variety and evaluated their models on a relatively narrow range of financial tasks, which is even fewer than some previous studies [1]. This raises concerns about the novelty and comprehensiveness of their work. It could further undermine the soundness of their training approach. Could you please elaborate on the advancements made by your research?

[1] Xie, Q., Li, D., Xiao, M., Jiang, Z., Xiang, R., Zhang, X., ... & Ananiadou, S. (2024). Open-finllms: Open multimodal large language models for financial applications. arXiv preprint arXiv:2408.11878.

3. The model comparison is based solely on various types of accuracy metrics. The paper lacks detailed statements regarding repeated experiments and does not provide error bars or significance levels for most of the experiments. Given that some accuracy results, such as those in Table 2 comparing their fine-tuned Phi-3 small with other models, are closely matched, it would be beneficial to include such evaluation metrics to ensure a more robust analysis.

**Questions:**

Please refer to the question proposed in the weaknesses section.

---

> ### Author Response · Authors · 2024-11-15
> **Response to Reviewer KnZt**
>
> We thank the reviewer for appreciating our presentation and for their insightful comments.  We address all points below.
>
> **Weakness I**: First, regarding the mention of '200+ models,' we acknowledge that this terminology may have been unclear without additional context. In this work, we refer to the checkpoint produced by each training experiment as a 'model.' However, we only use four base models, as described in Section 4.1: Phi-3-small, Phi-3-mini, Llama-3-8B, and Mistral-7B-v3. Thus, the 200+ models represent fine-tuned versions of these four base models. Based on a similar comment from Reviewer d8tj, we have added a clear explanation of the number of distinct training experiments that resulted in these fine-tuned models. For completeness, we repeat our response to Reviewer d8tj here:
>
> > Specifically, for each base LLM, we conducted 55 training experiments, and comparisons are made only within this subset. These 55 experiments per base LLM arise from the nine datasets, following the process outlined in Section 2.2. As shown, the number of training runs is calculated as $\binom{n}{1} + \binom{n}{2} + \binom{n-1}{1} + \binom{n}{n} = 55$, where $n=9$. We have added a concise explanation of this calculation to Section 4.1 in the manuscript.
>
> **Weakness II**: Regarding the reviewer's concern about the dataset variety: we acknowledge that the financial domain is vast, encompassing many intricate details that are not fully captured in our selection of seven core training datasets. However, we believe that these seven datasets provide a robust representation of critical financial tasks, including sentiment analysis, financial classification, named entity recognition, conversational capabilities, and tabular and mathematical understanding. These tasks collectively cover a substantial subset of financial NLP challenges.
>
> We also want to highlight the computational considerations inherent in this empirical study. Adding more datasets would significantly increase the computational demands due to the combinatorial growth in the number of training runs (as visualized in Figure 2). Given this trade-off, we carefully selected seven datasets that balance computational feasibility with a representative scope of financial tasks. Additionally, we have added a new limitations subsection to the manuscript to explicitly discuss this point.
>
> Regarding novelty and comprehensiveness, we agree that the work by Xie et al. is indeed relevant, and we note that our study shares some datasets with theirs. We have added a citation to their work in acknowledgment. However, our research makes a unique contribution by systematically evaluating the effects of dataset combinations on fine-tuning performance. Unlike prior studies, which often focus on fine-tuning a single model or evaluating pre-trained models, our work explores how the choice of training data mix can influence downstream performance. This emphasis on data selection as a critical variable distinguishes our study and offers valuable insights into fine-tuning strategies, particularly for financial NLP tasks, and potentially for the broader NLP community.
>
> **Weakness III**:  We thank the reviewer for suggesting the inclusion of error bars. In response, we have added margin of error values to Table 2, as recommended, to highlight significance. Additionally, we took the opportunity to enhance Figure 1 by including error bars for greater clarity.

---

> > ### Comment · Reviewer_KnZt · 2024-11-20
> >
> > Thanks for the authors' reply.
> >
> > In response to Weakness 1, based on the details provided, I find it difficult to consider the same model at different training checkpoints as distinct models. However, you highlight this as a major claim in several prominent sections of your manuscript to support the comprehensiveness of your approach. From my perspective, your explanation of the training process somewhat undermines the credibility of your final outcomes.
> >
> > Therefore, I decide to maintain my current rating.

---

> > > ### Author Response · Authors · 2024-11-26
> > >
> > > We thank the reviewer for their follow-up.
> > >
> > > In response to their comment, we have updated the manuscript to replace "over 200 models" with "over 200 training experiments/runs" where applicable, to more accurately describe our approach. However, we respectfully disagree with the assertion that using the same model at different training checkpoints—each representing **entirely separate training experiments**—undermines the comprehensiveness of our analysis. We also wish to clarify that our study does not evaluate different checkpoints from a single training process, but instead focuses on **independent** training runs conducted on distinct datasets.
> > >
> > > The core strength of our methodology lies in the exhaustive training across all dataset permutations, rather than the number of distinct base models used. The trends observed across four widely used open-sourced LLMs further reinforce the generality and validity of our findings. Moreover, the notion that a model’s behavior changes fundamentally across different training runs on different datasets is precisely the reason such an analysis is essential.
> > >
> > > We fail to see how this impacts the comprehensiveness of our approach, nor what additional insights would be gained by introducing yet another base model, given the consistent trends already demonstrated across the models presented in the manuscript, but we'd be happy to learn otherwise.

---

> > > > ### Comment · Reviewer_KnZt · 2024-11-28
> > > >
> > > > Thank you for the authors' clarification. I appreciate the effort put into the experiments. However, I believe that the choice of models and fine-tuning approaches lacks sufficient novelty and comprehensiveness to warrant a higher rating. As such, I have decided to maintain the current rating.

---

### Official Review · Reviewer_d8tJ · 2024-11-05

**Soundness:** 2
**Presentation:** 3
**Contribution:** 2
**Rating:** 5
**Confidence:** 4

**Summary:**

This paper presents extensive empirical results comparing the performance of various LLMs fine-tuned for a specific task to the performance of these LLMs fine-tuned on various combinations of tasks. The paper suggests that there often exists a combination of tasks that can improve performance on a downstream task of interest while also outperforming frontier models such as GPT-4o.

**Strengths:**

If code and all models are released, it could be a valuable resource for researchers studying topics such as data selection and model merging.

**Weaknesses:**

- It seems unfair to compare best-of-200 models evaluated on the test set vs a single model when arguing that multi-task training is beneficial. A comparable budget should be given to the single-task fine-tuning. You could, for example, perform fine-tuning for a single task with varying number of epochs (or using early stopping), learning rate, data shuffling, etc., to have an equivalent of 200 models and then report the best test performance.
- There is very little insight into how one should select the tasks for fine-tuning (besides that general instruction tuning with Orca datasets seems to help). There are also prior works that study similar problems which would be interesting to consider:

[1] Data Selection for Language Models via Importance Resampling. Xie et. al, 2023

[2] LESS: Selecting Influential Data for Targeted Instruction Tuning. Xia et. al, 2024

**Questions:**

When comparing to GPT-4o, did you verify that the gains of the smaller fine-tuned models are due to improved performance or because they follow the format better? GPT-4o may provide correct responses in the wrong format, leading to lower performance when using rule-based evaluation. Using LLM judge as you did for FinanceBench, could help to alleviate potential biases of rule-based evaluation (i.e. the judge compares LLM response to ground truth).

---

> ### Author Response · Authors · 2024-11-15
> **Response to Reviewer d8tJ**
>
> We thank the reviewer for raising valuable points to improve our work. We address all points below.
>
> **Open source**: In response to the reviewer's recommendation, we have open-sourced our project, including the source code and all datasets, accessible at https://anonymous.4open.science/r/cocktail_effect-54F8/README.md. While it is not feasible to upload all 200 saved models due to size limitations, the provided code enables full reproduction of our experiments or any chosen subset. Also, we'd be happy so share our trained models with researchers by uploading a subset of the trained models, upon request.
>
> **Weakness I**: Regarding the first weakness described, we appreciate the reviewer's focus on ensuring a fair comparison between the trained models, as this is indeed a key consideration. We assert that our comparison is fair for several reasons:
>
> - First, we clarify that we are not comparing a single model to a 'best-of-200' model. We recognize that our original language may have been unclear, so we have adjusted the wording in the paper to improve clarity. Specifically, for each base LLM, we conducted 55 training experiments, and comparisons are made only within this subset. These 55 experiments per base LLM arise from the nine datasets, following the process outlined in Section 2.2. As shown, the number of training runs is calculated as $\binom{n}{1} + \binom{n}{2} + \binom{n-1}{1} + \binom{n}{n} = 55$, where $n=9$. We have added a concise explanation of this calculation to Section 4.1 in the manuscript.
>
> - Second, regarding hyperparameter selection, we tested several configurations to find one that achieved good convergence across the various training runs. Given this hyperparameter search, additional tuning would likely not significantly impact the results, as this process has already been conducted. However, due to the large scale of experiments in this paper, a truly exhaustive search was not feasible, and we have added this as a limitation.
>
> - Lastly, our primary goal was to identify the optimal dataset mix, making it essential to train models on all possible combinations.
>
> **Weakness II**: Regarding the second weakness the reviewer has raised, we would like to thank the reviewer for pointing out these relevant papers, which we have now added acknowledgments to in our paper. We would also like to add that besides the insight the reviewer has mentioned, namely that using general instruction datasets and/or mathematical reasoning datasets is greatly helpful, we generally do not get to pick which tasks to include. The tasks are a given target set, based on the specific domain. Once this set has been established, the task of finding the optimal subset to train on (per task) is similar to any other hyperparameter search, where empirical experiments are necessary. Tables 5,6,7 in the appendix demonstrate the often non-intuitive nature of optimal combinations.
>
> **Question I**: The reviewer raises an important consideration regarding format adherence. To address this concern, we conducted a detailed empirical evaluation of the instruction-following capabilities of GPT-4-o within our benchmarks. Specifically, we identified all instances where our rule-based evaluation failed to extract an answer, which directly corresponds to cases where GPT-4-o did not correctly follow the required format. The results, presented in Table 1 below, show that the number of formatting issues is negligible, thus reducing the need to use LLM-as-a-judge in this case. Furthermore, this demonstrates that the gains of the smaller fine-tuned models are due to improved performance relative to GPT-4-o, rather than being limited to better format adherence.
>
>
> **Table 1: Number of GPT-4-o formatting errors per task, presented as a fraction (errors/total test set samples).**
> | **Dataset**       | **# GPT-4-o Formatting Mistakes** |
> |--------------------|-----------------------------------|
> | Twitter SA         | 3/2,388                          |
> | TwitterTopics      | 2/4,117                          |
> | FinNerCLS          | 1/3,502                          |
> | FPB                | 0/970                            |
> | ConvFinQA          | 0/1,486                          |
> | FinQA              | 0/1,125                          |
> | Headline           | 0/20,547                         |

---

### Author Response · Authors · 2024-11-15
**Updated Manuscript**

We would like to thank all reviewers for their valuable feedback and thoughtful suggestions to improve our paper. We have addressed all points in our official comments and updated the manuscript PDF accordingly. The main changes include:

- Adding error bars to Figure 1 and margin of error values to Table 2.
- Incorporating relevant citations suggested by the reviewers.
- Clarifying the experimental setup in Section 4.1.
- Adding a dedicated limitations section to the paper.

We appreciate your insightful feedback and believe these changes strengthen the manuscript. We look forward to receiving your feedback and continuing our discussions with each of you.

---

### Note · Authors · 2024-12-03

**Comment:**

We would like to thank the reviewers for their thoughtful feedback during the initial review stage, which significantly helped us refine our work. While we made every effort to address their suggestions in our revised manuscript, we found the discussion phase less productive than we had hoped, as delays and limited responses constrained meaningful engagement.

After careful consideration, we have decided to withdraw our paper out of respect for the Area Chair's time. We sincerely thank the reviewers and Area Chair for their time and effort throughout the review process.

**Withdrawal Confirmation:**

I have read and agree with the venue's withdrawal policy on behalf of myself and my co-authors.